# Real-world treatment trends and triple class exposed status in newly diagnosed multiple myeloma patients in Japan: A retrospective claims database study

Toyoki Moribe[1][☯]*, Linghua Xu[2][☯], Kazumi Take[1], Naohiro Yonemoto[2], Kenshi Suzuki[3]

1 Oncology Medical Affairs, Pfizer Japan Inc., Tokyo, Japan, 2 Access & Value, Pfizer Japan Inc., Tokyo, Japan, 3 Department of Hematology, Japanese Red Cross Medical Center, Tokyo, Japan

☯ These authors contributed equally to this work.
* tmoribe1025@gmail.com

**Data Availability Statement:** In this study, we used the data set provided from a third party, Medical Data Vision Co., Ltd (MDV) (Tokyo,

## Abstract

Treatment trends for newly diagnosed multiple myeloma (NDMM) are not fully evaluated in real-world settings in the Japanese population. Triple-class exposed (TCE) patients with relapsed or refractory MM have a poor prognosis and limited treatment options. To clarify characteristics, treatment trends, and TCE status in Japanese patients with MM, we conducted a retrospective, non-interventional study. Data from patients with MM were extracted from a Japanese claims database between 2015 and 2022: this study identified patients with NDMM prescribed daratumumab (D), lenalidomide (R), and/or bortezomib (V) as 1st-line treatment. The patient characteristics and treatment trends were analyzed for non-transplant and transplant groups. Of 1,784 patients, non-transplant patients (n = 1,656, median age 75 years [range: 37–94]) received R+dexamethasone (Rd) (24.7%), Vd (23.8%), and RVd (15.6%) and transplant patients (n = 128, median age 61 years [range: 35–73]) received RVd (49.5%), Vd (18.7%), and DVd (8.4%) in 1st line. In the non-transplant group, the commonly prescribed treatment regimens were Rd for patients aged ≥75 years, Vd for patients aged 65–74 years, and RVd for patients aged <65 years. Patients with renal or cardiac dysfunction commonly received Vd or Rd, respectively. In the transplant group, 107 (83.6%) and 20 (15.6%) patients received transplantation in the 1st and 2nd lines, respectively. The top three regimens as induction therapy before stem cell transplantation were RVd (49.5%), Vd (18.7%), or DVd (8.4%) in 1st line. Cumulative TCE patients by 5th line were 351 (21.2%) and 56 (43.8%) for non-transplant and transplant patients, respectively. TCE ratio at each line gradually increased from 1st to 5th line (11.1–69.2% in the non-transplant group and 21.1–100% in the transplant group, respectively). Of 184 TCE patients in the non-transplant group, 89.7% received sequencing treatments including DRd, RVd, and DVd, and 10.3% received D-RVd in 1st line.

Japan). Authors cannot receive any rights in accessing the data set and sharing it with other researchers. Any other researchers would need to apply to gain permission to access and use the data set from the third party, Medical Data Vision Co., Ltd. For inquiries on access to the dataset used in this study, please contact MDV (website: (JP) https://www.mdv.co.jp/ (ENG) https://en.mdv.co.jp/; e-mail: ebm_sales@mdv.co.jp).

**Funding:** The funder, Pfizer Japan Inc., provided support in the form of salaries for authors [TM, LX, KT and NY]. The medical writing support was also funded by Pfizer Japan Inc. (Tokyo, Japan). The funders had no additional role in study design, data collection and analysis, decision to publish, or preparation of the manuscript. The specific roles of these authors are articulated in the 'author contributions' section.

**Competing interests:** Toyoki Moribe, Linghua Xu, Kazumi Take, and Naohiro Yonemoto are employees of Pfizer Japan Inc., a sponsor of this study. Statistical analysis support was provided by Tatsuo Sakashita and Takayuki Sawada of Clinical Study Support, Inc. (Nagoya, Japan), which was funded by Pfizer Japan Inc. There are no patents, products in development or marketed products associated with this research to declare. This commercial affiliation, Pfizer Japan Inc., does not alter our adherence to PLOS ONE policies on sharing data and materials.

## Introduction

Multiple myeloma (MM) is marked by the uncontrolled growth of monoclonal plasma cells in the bone marrow resulting in the production of dysfunctional immunoglobulins leading to substantial morbidity and mortality. As per the latest Global Cancer Observatory statistics, there were an estimated 176,404 patients with MM globally in 2020, accounting for 0.91% of all cancer diagnoses and about 1.1% of deaths. The 5-year prevalence was 5.78 per 100,000 population. Global age-standardized incidence rate of MM was 1.8 per 100,000 persons in 2020 [1]. In Japan, the incidence of MM in the year 2019 was reported as approximately 5 per 100,000 persons per year resulting in 4,000 deaths per year, and the incidence and mortality rates have been increasing yearly [2]. Since MM is not yet curable, treatment strategies aim to extend the life expectancy of patients. Indeed, a systematic review conducted by Kumar et al. suggested that increasingly effective new treatment strategies and enhanced supportive care have led to improved survival for patients with MM based on response evaluation [3]. The initial treatment approach depends on the suitability of the patient for stem cell transplantation (SCT) following initial chemotherapy [4]. The Japanese Society of Hematology (JSH) Practical Guidelines for Hematological Malignancies consider patients aged <65 years and with normal major organ functioning eligible for SCT, while patients aged ≥65 years or suffering from organ dysfunction or any other immune-related risk factors, are considered ineligible [2,5,6].

Since the early 2000s, numerous treatment agents and regimens have emerged for treating MM. In Japan, bortezomib (approved in October 2006), thalidomide (approved in October 2008), and lenalidomide (approved in June 2010) were approved as treatment to prolong progression-free survival of patients with relapsed or refractory MM (RRMM). Additionally, bortezomib (approved in September 2011) and lenalidomide (approved in December 2015) became available for patients with newly diagnosed MM (NDMM) [4]. Furthermore, various treatment regimens including seven drugs (pomalidomide, carfilzomib, ixazomib, panobinostat, elotuzumab, daratumumab, and isatuximab) have been developed and prescribed to patients with NDMM or RRMM [4]. Of these, daratumumab has become a key drug for MM therapy. In Japan, daratumumab was approved for the indication of RRMM in September 2017, and daratumumab in combination with bortezomib, melphalan, and prednisone (D-VMP), and in combination with lenalidomide and dexamethasone (DRd) regimens were approved in 2019 for patients with NDMM who were ineligible for SCT. However, daratumumab combination regimens used in the 1st line have complicated MM treatment and fragmented later-line MM treatment sequences.

This has created a situation of increasing numbers of triple-class exposed (TCE) patients who are treated with immunomodulatory drugs (IMiDs), proteasome inhibitors (PIs), and anti-CD38 monoclonal antibodies. TCE patients have limited treatment options, especially before chimeric antigen receptor T-cell (CAR-T) therapies and bispecific antibodies (BsAbs) become available. At present, there are two United States Food and Drug Administration–approved CAR-T products for the treatment of RRMM (idecaptagene vicleucel [ide-cel] and ciltacabtagene autoleucel [cilta-cel]). Three T-cell engaging BsAbs (teclistamab, elranatamab, and talquetamab) have also received approval in the USA for use in previously treated patients with MM. In Japan, elranatamab has recently been approved for TCE patients with RRMM. Of these treatment options, only one CAR-T therapy (ide-cel, Abecma®, Bristol Myers Squibb) has been approved and integrated into the market for use in Japan. Despite these therapies, patients tend to relapse or become refractory over time. TCE patients who have relapsed or are refractory have a poor prognosis with worsening outcomes and limited treatment options. Therefore, it is essential to identify the current real-world TCE patient population in each line for non-transplant and transplant patients with NDMM to establish a clinical consensus on

standards of care and treatment strategies for TCE patients, especially in an RRMM setting, in Japan.

The treatment patterns and sequences adopted for treating NDMM and RRMM are not completely evaluated in real-world settings in the Japanese population on a large scale. Some previous real-world treatment studies have reported that patients utilize treatment regimens comprising chemotherapy, immunotherapy, and SCT. These studies have provided information on the presence/absence of transplantation, treatment costs, initiation time-to-treatment, regimen selection by age, and the duration of treatment in Japanese patients with MM [4,5,7,8]. In addition, patient characteristics, treatment patterns, and healthcare costs for real-world TCE patients with RRMM have been reported in Japan [9]. As these studies are retrospective in nature using various claims databases and data from very limited numbers of patients [5–7,9,10], there is a need for a more realistic picture of treatment approaches for NDMM in more hospitals and a greater number of patients.

We conducted a retrospective, non-interventional study using a large claims database to describe the patient characteristics, treatment trends, TCE patient status, and treatment sequences in each treatment line among non-transplant and transplant patients with NDMM who were prescribed daratumumab, lenalidomide, and/or bortezomib as 1st-line treatment between January 2015 and December 2022 in clinical practice in Japan. Furthermore, we also addressed the preferred regimens by age and comorbidity in non-transplant patients with NDMM.

## Materials and methods

### Study design and data source

This is a retrospective, observational study of patients with NDMM who were prescribed bortezomib and/or lenalidomide and/or daratumumab as 1st-line treatment regimens using hospital-based administrative claims data from Japanese hospitals, compiled by Medical Data Vision (MDV) Co., Ltd (Tokyo, Japan) (S1 Fig). The MDV database is the largest commercially available administrative claims database in Japan and includes data of approximately 44 million patients (including a substantial proportion of those aged >65 years) from over 480 hospitals across Japan. It contains inpatient and outpatient hospitals, and prescription data collected after a hospital visit, as well as health claims data [11]. The MDV database has been widely used in a variety of real-world database studies in Japan [12].

### Study population and study period

In the study period from January 1, 2015, to December 31, 2022, patients who visited healthcare facilities registered with the MDV database, had at least one record of MM identified using the International Statistical Classification of Diseases and Related Health Problems, 10th Revision (ICD10; C90: Multiple myeloma and malignant plasma cell neoplasms), had ≥6 months follow-up data, and were prescribed bortezomib, lenalidomide, and/or daratumumab after the first diagnosis of MM were included in the study. Therefore, the results do not include the data of treated patients with an initial diagnosis before 2015. The index date was defined as the earliest prescription date of the 1st-line drugs in the same month or months after the first diagnosis of MM. Patients were excluded from the study if they were prescribed anti-myeloma treatment drugs other than dexamethasone and prednisone, were subjected to SCT between April 1, 2008, and the day before the index date, or were without any medical record for ≥3 years before the index date. The study population was grouped as non-transplant and transplant patients based on whether they received SCT or not after the index date.

The baseline period was considered to be 6 months before the index date, while the end of follow-up was defined as the date of the last identified data record in the MDV database after the index date.

## Variables

Baseline characteristics including age, age category, and sex at index date; comorbidities; and comorbidity class at baseline were assessed. Comorbidities were defined using the ICD10 classification recorded within 6 months before the index date (S1 Table). Treatment characteristics such as hospital size (number of beds), hospital category, status of autologous SCT, and drug treatment patterns (type of drug, type/line of treatment regimen, prescription period based on the start and end date of each oral drug, date of administration of injectable drugs, start and end date of each treatment, line and duration of each treatment) were also assessed.

The treatment regimen is defined as a combination of different classes of anti-myeloma drugs (an IMiD, an anti-CD38 antibody drug, a PI, an anti-signaling lymphocytic activation molecule family 7 antibody, and/or a histone deacetylase inhibitor), melphalan, dexamethasone, and prednisone prescribed during the first 56-day period after the earliest prescription date of each MM therapy. Double-class and triple-class drug regimens described in Table 1 were also evaluated. In some cases, the identified regimen during the first 56-day period was not changed in a treatment line. In other cases, a drug was added into the regimen, a drug or drugs was discontinued, or the regimen was sequentially bridged to another regimen in a treatment line. The regimens were calculated together as the regimen based in our study.

**Table 1. Treatment regimens described in the study.**

| Treatment regimen | Drug name(s) |
| --- | --- |
| Bor-based | Bortezomib (V) |
| Dara-based | Daratumumab (D) |
| DKd-based | Daratumumab (D), carfilzomib (K), dexamethasone (d) |
| DRd-based | Daratumumab (D), lenalidomide (R), dexamethasone (d) |
| DVd-based | Daratumumab (D), bortezomib (V), dexamethasone (d) |
| D-VMP-based | Daratumumab (D), bortezomib (V), melphalan (M), prednisone |
| EPd-based | Elotuzumab (E), pomalidomide (P), dexamethasone (d) |
| ERd-based | Elotuzumab (E), lenalidomide (R), dexamethasone (d) |
| FVd-based | Panobinostat (F), bortezomib (V), dexamethasone (d) |
| IRd-based | Ixazomib (I), lenalidomide (R), dexamethasone (d) |
| Ixa-based | Ixazomib (Ixa) |
| Kd-based | Carfilzomib (K), dexamethasone (d) |
| KRd-based | Carfilzomib (K), lenalidomide (R), dexamethasone (d) |
| Len-based | Lenalidomide (Len) |
| Pd-based | Pomalidomide (P), dexamethasone (d) |
| PVd-based | Pomalidomide(P), bortezomib (V), dexamethasone (d) |
| Rd-based | Lenalidomide (R), dexamethasone (d) |
| RVd-based | Lenalidomide (R), bortezomib (V), dexamethasone (d) |
| Sar-based | Isatuximab (Sar) |
| Sd-based | Isatuximab (S), dexamethasone (d) |
| SPd-based | Isatuximab (S), pomalidomide (P), dexamethasone (d) |
| Td-based | Thalidomide (T), dexamethasone (d) |
| Vd-based | Bortezomib (V), dexamethasone (d) |
| VMP-based | Bortezomib (V), melphalan (M), prednisone |

Treatment line was defined as the use of the current class of anti-myeloma drugs either singly or in combination, from the earliest prescription date to the last date of the current prescription. Treatment lines were characterized for non-transplant and transplant patients according to the drugs prescribed in the current and subsequent regimens as well as the time (days) to the start of the next drug therapy. More details on treatment lines, sequences, and line transfer are described in S2 and S3 Figs.

A specific outcome variable of interest was TCE status in each line. TCE patients were defined as those who received at least one IMiD, at least one PI, and at least one anti-CD38 monoclonal antibody in this study.

### Statistical analysis

In this study, we have presented the data in a descriptive manner, overall and by transplant history (eligibility) after the index date. The analyses were conducted separately for the non-transplant and transplant groups. For categorical variables (e.g., patient characteristics [age, age category, sex, comorbidities, hospital size, and category]), frequency (n) and percentages of patients were calculated. For continuous variables (e.g., age, follow-up period), summary statistics (median, minimum, maximum, interquartile range [IQR]) were presented.

Treatment patterns and treatment lines (patients who experienced 1st-, 2nd-, 3rd-, 4th- or later-line treatment and the most recent treatment line) were presented as frequency and percentages. The percentages of patients in each treatment line were calculated by dividing the number of patients in a line of treatment by the total number of patients in the corresponding treatment line. Duration of treatment (months) was calculated by dividing the treatment duration in days by 30.4375 in each line of treatment. Treatment regimens were described by frequency and percentages of patients receiving each regimen during the treatment period. Treatment pattern was also described by the frequency and percentages of patients receiving the treatment pattern in each line and calculated in the same way as treatment regimen. Treatment lines and treatment regimens were also evaluated by age and comorbidities.

Variables in patients with NDMM in non-transplant and transplant groups were evaluated descriptively with summary statistics. The numbers of TCE patients were summarized by frequency and the percentages were calculated by dividing the number of TCE patients by the number of patients in each treatment line. All data analysis was performed using statistical software SAS version 9.4 (SAS Institute, Cary, NC, USA).

### Ethical approval and consent to participate

The study was conducted in accordance with legal and regulatory requirements, including data protection laws. This study is not applicable to the Japanese Government's "Ethical Guidelines for Medical and Health Research Involving Human Subjects" because this study used secondary data that are anonymized by a third party, Medical Data Vision Co., Ltd (Tokyo, Japan). Informed consent and ethics committee approval were not required.

## Results

### Study population

Over the study period, 44,897 patients had a confirmed diagnosis of MM, of which 16,021 patients had prescriptions for daratumumab, lenalidomide, and/or bortezomib at the index date (Fig 1). Of 12,255 patients who had a follow-up period of at least 6 months from the index date, 10,471 patients who met the other three exclusion criteria—were prescribed anti-myeloma treatment drugs other than dexamethasone and prednisone before the index date, had

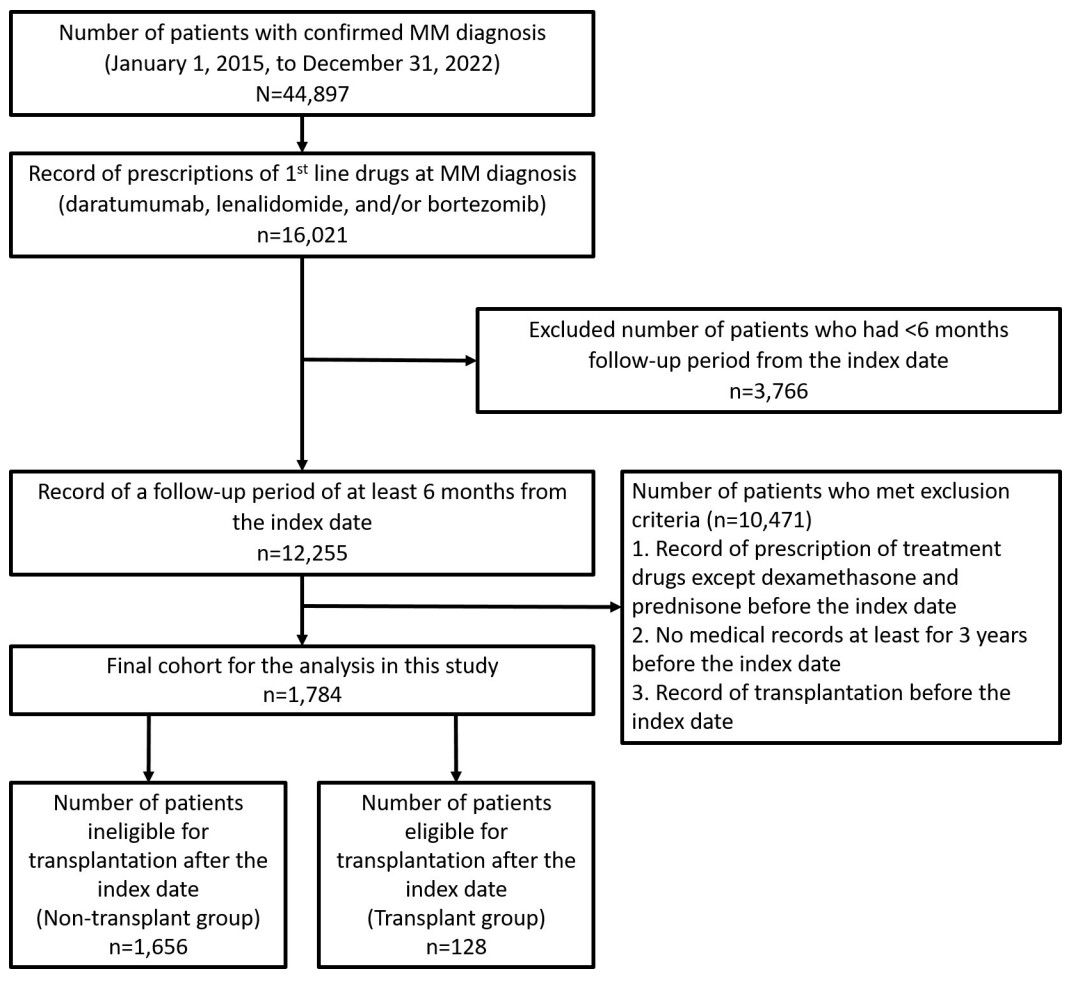

**Fig 1. Patient disposition.** MM, multiple myeloma.

no medical records before the index date for ≥3 years, and received SCT before the index date —were excluded. Thus, 1,784 patients were included in the final cohort analysis set. Of the 1,784 patients, 1,656 patients were transplant ineligible (non-transplant group) and 128 patients were transplant eligible (transplant group) (Fig 1).

### Patient characteristics

The median (minimum–maximum) age of patients was 75 (37–94) years and 61 (35–73) years in the non-transplant and transplant groups, respectively. Of 1,656 patients in the non-transplant group, the majority (77.6%) were aged 65 to 84 years while 10.7% were 18 to 64 years, and 11.7% were older than 85 years. Notably, more than half (55.4%) were older than 74 years. Of 128 patients in the transplant group, all patients were younger than 75 years (71.1% aged 18 to 64 years and 28.9% aged 65 to 74 years) but some had comorbidities including renal (7.0%), liver (4.7%), and cardiac (6.3%) dysfunction, vascular disorder (9.4%), and back pain (13.3%) during the baseline period. Contrastingly, of 1,656 patients in the non-transplant group, higher proportions of patients had different complex comorbidities including renal (16.2%), liver (6.8%), cardiac (24.3%), and pulmonary (3.9%) dysfunction, vascular disorder (32.5%), and back pain (15.9%) during the baseline period.

Amongst the study population of 1,784 patients, 49.8% and 39.8% patients visited hospitals with 200–499 beds while 47.5% and 60.2% patients visited hospitals with ≥500 beds in the non-transplant and transplant groups, respectively. Of 1,656 patients in the non-transplant group and 128 patients in the transplant group, 87.4% and 93.7% patients, respectively, visited public hospitals or university hospitals for the treatment of MM. In contrast, 17.6% patients in the non-transplant group and 6.3% patients in the transplant group received treatment at private hospitals (Table 2).

Specifically, among patients (1,152 and 76 patients in the non-transplant and transplant groups, respectively) whose diagnostic records for comorbidity existed during baseline period, 40.5% and 11.8% patients had primary hypertension, 21.8% and 2.6% had heart failure, 14.6%

**Table 2. Patient characteristics.**

| Characteristic | Non-transplant group (N = 1,656) | Transplant group (N = 128) |
|---|---|---|
| **Age, (years)** | | |
| Mean (SD) | 74.7 (8.8) | 59.4 (8.1) |
| Median (min–max) | 75.0 (37–94) | 61.0 (35–73) |
| Q1–Q3 | 70.0–81.0 | 54.0–65.5 |
| **Age category (years), n (%)** | | |
| <18 | 0 (0.0) | 0 (0.0) |
| 18–64 | 178 (10.7) | 91 (71.1) |
| 65–74 | 561 (33.9) | 37 (28.9) |
| 75–84 | 723 (43.7) | 0 (0.0) |
| ≥85 | 194 (11.7) | 0 (0.0) |
| **Sex, n (%)** | | |
| Male | 870 (52.5) | 62 (48.4) |
| Female | 786 (47.5) | 66 (51.6) |
| **Comorbidities, n (%)** | | |
| Renal dysfunction | 269 (16.2) | 9 (7.0) |
| Liver dysfunction | 112 (6.8) | 6 (4.7) |
| Cardiac dysfunction | 402 (24.3) | 8 (6.3) |
| Pulmonary dysfunction | 64 (3.9) | 0 (0.0) |
| Vascular disorder | 539 (32.5) | 12 (9.4) |
| Back pain | 263 (15.9) | 17 (13.3) |
| Dementia | 9 (0.5) | 0 (0.0) |
| **Hospital size, n (%)** | | |
| ≤199 beds | 46 (2.8) | 0 (0.0) |
| 200–499 beds | 824 (49.8) | 51 (39.8) |
| ≥500 beds | 786 (47.5) | 77 (60.2) |
| **Hospital category, n (%)** | | |
| University hospital | 82 (5.0) | 10 (7.8) |
| Public hospital | 1,365 (82.4) | 110 (85.9) |
| Private hospital | 209 (12.6) | 8 (6.3) |
| **Follow-up period (months)** | | |
| Mean (SD) | 28.0 (18.6) | 34.7 (19.7) |
| Median (min–max) | 23.0 (5.9–94.3) | 30.9 (7.7–95.2) |
| Q1–Q3 | 13.2–38.6 | 18.5–48.1 |

max, maximum; min, minimum; Q1, 1st quartile; Q3, 3rd quartile; SD, standard deviation

**Table 3. Major medical history of patients whose diagnostic records for comorbidity existed during baseline period.**

| ICD10 code | ICD10 name of disease | Non-transplant group (N = 1,152) | Transplant group (N = 76) |
|---|---|---|---|
| I10 | Essential (primary) hypertension | 466 (40.5) | 9 (11.8) |
| I50 | Heart failure | 251 (21.8) | 2 (2.6) |
| I20 | Angina pectoris | 168 (14.6) | 5 (6.6) |
| N18 | Chronic kidney diseases | 137 (11.9) | 3 (3.9) |
| I48 | Atrial fibrillation and atrial flutter, unspecified | 100 (8.7) | 0 (0.0) |

Data are presented as n (%).

ICD10, International Statistical Classification of Diseases and Related Health Problems, 10th Revision.

and 6.6% had angina pectoris, 11.9% and 3.9% had chronic kidney disease, and 8.7% and 0.0% had atrial fibrillation/flutter as medical history in the non-transplant and transplant groups, respectively (Table 3).

## Treatment line and duration of treatment

Median (minimum–maximum) follow-up duration was 23.0 (5.9–94.3) months and 30.9 (7.7–95.2) months in the non-transplant and transplant groups, respectively. During follow-up, 36.5%, 9.4%, and 2.7% patients in the non-transplant group and 59.4%, 20.3%, and 7.0% patients in the transplant group moved to 2nd, 3rd, and 4th or later lines of treatment, respectively (Table 4). Most recent therapies with 1st, 2nd, 3rd, and 4th or later treatment lines were received by 63.5%, 27.1%, 6.7%, and 2.7% patients in the non-transplant group and 40.6%, 39.1%, 13.3%, and 7.0% patients in the transplant group, respectively (Table 4).

Duration of treatment tended to be slightly shorter in the non-transplant group than the transplant group: 10.86, 7.56, 4.48, and 5.29 months in the non-transplant group and 11.76, 8.08, 8.23, and 4.47 months in the transplant group for the 1st, 2nd, 3rd, and 4th line, respectively. For each treatment line, the duration of treatment was shorter in those who received a

**Table 4. Treatment lines.**

| Treatment line | Non-transplant group (N = 1,656) | Transplant group (N = 128) |
|---|---|---|
| **(a) Cumulative number and proportion of patients using therapies across different treatment lines** | | |
| 1st line | 1,656 (100.0) | 128 (100.0) |
| 2nd line | 605 (36.5) | 76 (59.4) |
| 3rd line | 156 (9.4) | 26 (20.3) |
| 4th line or more | 45 (2.7) | 9 (7.0) |
| **(b) Number and proportion of patients based on the most recent treatment lines being used** | | |
| 1st line | 1,051 (63.5) | 52 (40.6) |
| 2nd line | 449 (27.1) | 50 (39.1) |
| 3rd line | 111 (6.7) | 17 (13.3) |
| 4th line or more | 45 (2.7) | 9 (7.0) |

Data are presented as n (%).

"Cumulative number and proportion of patients using therapies across different treatment lines" represents the total number of patients who have moved to each treatment line and experienced treatment in that line. "Number and proportion of patients based on the most recent treatment lines being used" represents the patients who are currently receiving treatment in each treatment line.

**Table 5. Duration of treatment in non-transplant and transplant groups.**

| Treatment line | Duration of treatment (months) | | | | | |
| --- | --- | --- | --- | --- | --- | --- |
| | Non-transplant group | | | Transplant group | | |
| | Overall | Subsequent line treatment | | Overall | Subsequent line treatment | |
| | | Prescribed | Not prescribed | | Prescribed | Not prescribed |
| | N = 1,656 | n = 605 | n = 1,051 | N = 128 | n = 76 | n = 52 |
| 1st line | 10.86 (0.03–95.08) | 6.90 (0.03–70.67) | 14.32 (0.03–95.08) | 11.76 (0.43–76.06) | 4.80 (0.43–66.60) | 18.96 (4.70–76.06) |
| 2nd line | 7.56 (0.03–82.46) | 6.46 (0.03–44.85) | 8.51 (0.03–82.46) | 8.08 (0.10–67.12) | 2.92 (0.69–56.21) | 9.91 (0.10–67.12) |
| 3rd line | 4.48 (0.03–66.60) | 4.07 (0.07–40.67) | 4.76 (0.03–66.60) | 8.23 (0.46–47.77) | 4.01 (0.46–15.80) | 10.02 (1.97–47.77) |
| 4th line | 5.29 (0.03–43.89) | 3.45 (0.23–15.61) | 7.00 (0.03–43.89) | 4.47 (0.10–30.55) | 3.25 (0.72–6.44) | 7.54 (0.10–30.55) |

Duration of treatment in months expressed as median (minimum–-maximum).

subsequent line of therapy than in those who did not. Duration of treatment was longest (18.96 months) for patients in the 1st line in the transplant group and 14.32 months in the 1st line in the non-transplant group when a subsequent line of treatment was not prescribed (Table 5).

## Frequently used treatment regimens

Of 1,656 non-transplant patients, 24.7% received lenalidomide and dexamethasone (Rd)-based 1st-line treatment, 23.8% received bortezomib and dexamethasone (Vd)-based, and 15.6% received lenalidomide, bortezomib, and dexamethasone (RVd)-based therapies: for 2nd-line treatment, the corresponding proportions were 22.0%, 9.8%, and 3.6%, respectively (Table 6 and Fig 2). Together, the Rd-, Vd-, or RVd-based therapies were used to treat 64.1%, 35.4%, 21.1%, and 13.3% patients in 1st-, 2nd-, 3rd-, and 4th-line treatments, respectively. Daratumumab-containing regimens (daratumumab, bortezomib, melphalan, and prednisone [D-VMP]; daratumumab, lenalidomide, and dexamethasone [DRd]; daratumumab, bortezomib, and dexamethasone [DVd]; daratumumab, carfilzomib, and dexamethasone [DKd]; or daratumumab-based therapies) were prescribed to 10.0%, 16.2%, 10.9%, and 20.0% patients in 1st-, 2nd-, 3rd-, and 4th-line treatments, respectively.

Of 128 patients in the transplant group, 107 (83.6%), 20 (15.6%), and 1 (0.8%) received transplantation in the 1st, 2nd, and 4th lines, respectively. The three most frequently prescribed regimens for 1st-line induction therapy before SCT were RVd-based (49.5%), Vd-based (18.7%), and DVd-based (8.4%). For 2nd-line induction therapy, bortezomib-based (25.0%), Rd-based (20.0%), and carfilzomib, lenalidomide, and dexamethasone (KRd)-based or carfilzomib and dexamethasone (Kd)-based (each 10.0%) were prescribed most frequently (Table 6 and Fig 2). Among 107 patients who received transplantation in the 1st line, 25.5%, 6.7%, and 14.3% patients received Rd-based therapy in 2nd, 3rd, and 4th treatment lines, respectively. Lenalidomide-based treatment was also prescribed to 16.4%, 13.3%, and 14.3% patients in 2nd, 3rd, and 4th line treatments, respectively. Together, Rd- or lenalidomide-based therapies were used to treat 41.8%, 20.0%, and 28.6% patients in 2nd, 3rd, and 4th treatment lines, respectively. An RVd-based regimen was not used in the 2nd or later lines for transplant group patients (Table 6 and Fig 2).

In the non-transplant group, the commonly prescribed treatment regimens were Rd-based for patients aged ≥75 years (28.2%, 75–84 years; 43.3%, ≥85 years), Vd-based for patients

**Table 6. Treatment regimen in each line in non-transplant and transplant groups.**

| Treatment regimen | 1st line | 2nd line | 3rd line | 4th line |
|---|---|---|---|---|
| **(a) Non-transplant group** | **(n = 1,656)** | **(n = 605)** | **(n = 156)** | **(n = 45)** |
| Rd-based | 409 (24.7) | 133 (22.0) | 23 (14.7) | 3 (6.7) |
| Vd-based | 394 (23.8) | 59 (9.8) | 9 (5.8) | 1 (2.2) |
| RVd-based | 259 (15.6) | 22 (3.6) | 1 (0.6) | 2 (4.4) |
| DRd-based | 110 (6.6) | 52 (8.6) | 6 (3.8) | 3 (6.7) |
| Bor-based | 65 (3.9) | 24 (4.0) | 3 (1.9) | 1 (2.2) |
| DVd-based | 49 (3.0) | 20 (3.3) | 2 (1.3) | 3 (6.7) |
| VMP-based[a] | 47 (2.8) | 4 (0.7) | 2 (1.3) | 0 (0.0) |
| Len-based | 43 (2.6) | 42 (6.9) | 12 (7.7) | 2 (4.4) |
| IRd-based | 13 (0.8) | 16 (2.6) | 5 (3.2) | 2 (4.4) |
| D-VMP-based[a] | 4 (0.2) | 0 (0.0) | 0 (0.0) | 1 (2.2) |
| ERd-based | 4 (0.2) | 4 (0.7) | 3 (1.9) | 3 (6.7) |
| KRd-based | 4 (0.2) | 5 (0.8) | 3 (1.9) | 0 (0.0) |
| Dara-based | 2 (0.1) | 18 (3.0) | 6 (3.8) | 1 (2.2) |
| PVd-based | 2 (0.1) | 3 (0.5) | 0 (0.0) | 0 (0.0) |
| DKd-based | 1 (0.1) | 8 (1.3) | 3 (1.9) | 1 (2.2) |
| EPd-based | 0 (0.0) | 3 (0.5) | 3 (1.9 | 4 (8.9) |
| FVd-based | 0 (0.0) | 0 (0.0) | 1 (0.6) | 1 (2.2) |
| Kd-based | 0 (0.0) | 12 (2.0) | 11 (7.1) | 2 (4.4) |
| Pd-based | 0 (0.0) | 23 (3.8) | 11 (7.1) | 1 (2.2) |
| SPd-based | 0 (0.0) | 9 (1.5) | 1 (0.6) | 1 (2.2) |
| Ixa-based | 0 (0.0) | 15 (2.5) | 3 (1.9) | 0 (0.0) |
| Sar-based | 0 (0.0) | 1 (0.2) | 1 (0.6) | 0 (0.0) |
| Sd-based | 0 (0.0) | 0 (0.0) | 1 (0.6) | 0 (0.0) |
| Td-based | 0 (0.0) | 1 (0.2) | 0 (0.0) | 0 (0.0) |
| Other | 250 (15.1) | 131 (21.7) | 46 (29.5) | 13 (28.9) |
| **(b) Transplant group** | **(n = 107)** | **(n = 55)** | **(n = 15)** | **(n = 7)** |
| RVd-based | 53 (49.5) | 0 (0.0) | 0 (0.0) | 0 (0.0) |
| Vd-based | 20 (18.7) | 1 (1.8) | 1 (6.7) | 0 (0.0) |
| DVd-based | 9 (8.4) | 2 (3.6) | 1 (6.7) | 0 (0.0) |
| DRd-based | 4 (3.7) | 5 (9.1) | 0 (0.0) | 0 (0.0) |
| Rd-based | 3 (2.8) | 14 (25.5) | 1 (6.7) | 1 (14.3) |
| Bor-based | 2 (1.9) | 1 (1.8) | 0 (0.0) | 0 (0.0) |
| KRd-based | 1 (0.9) | 1 (1.8) | 1 (6.7) | 0 (0.0) |
| Len-based | 0 (0.0) | 9 (16.4) | 2 (13.3) | 1 (14.3) |
| Ixa-based | 0 (0.0) | 5 (9.1) | 0 (0.0) | 0 (0.0) |
| ERd-based | 0 (0.0) | 2 (3.6) | 1 (6.7) | 0 (0.0) |
| SPd-based | 0 (0.0) | 5 (9.1) | 0 (0.0) | 0 (0.0) |
| Dara-based | 0 (0.0) | 1 (1.8) | 1 (6.7) | 0 (0.0) |
| Kd-based | 0 (0.0) | 1 (1.8) | 1 (6.7) | 0 (0.0) |
| Pd-based | 0 (0.0) | 1 (1.8) | 1 (6.7) | 1 (14.3) |
| EPd-based | 0 (0.0) | 0 (0.0) | 1 (6.7) | 0 (0.0) |
| IRd-based | 0 (0.0) | 0 (0.0) | 0 (0.0) | 1 (14.3) |
| SKd-based | 0 (0.0) | 0 (0.0) | 0 (0.0) | 1 (14.3) |
| Sd-based | 0 (0.0) | 0 (0.0) | 0 (0.0) | 1 (14.3) |

(*Continued*)

**Table 6.** (Continued)

| Treatment regimen | 1st line | 2nd line | 3rd line | 4th line |
|---|---|---|---|---|
| Other | 15 (14.0) | 7 (12.7) | 4 (26.7) | 1 (14.3) |

Data are presented as n (%).

[a]P represents prednisone.

d, dexamethasone; D, daratumumab; E, elotuzumab; F, panobinostat; I, ixazomib; Ixa, ixazomib; K, carfilzomib; len, lenalidomide; M, melphalan; P, pomalidomide; R, lenalidomide; S, isatuximab: Sar, isatuximab; T, thalidomide; V, bortezomib.

aged 65–74 years (24.6%), and RVd-based for patients aged <65 years (32.0%) (S2 Table). In the non-transplant group, the most common regimens for patients with renal dysfunction and vascular disorder were Vd-based, for those with cardiac or pulmonary dysfunction Rd-based, and for those with liver dysfunction Rd-based and Vd-based equally (S3 Table).

## TCE patients

Cumulative TCE patients and TCE patients per line are summarized in Table 7 and Fig 3. In the non-transplant group, among patients receiving 1st, 2nd, 3rd, 4th, and 5th or later lines of treatments, 11.1%, 29.6%, 39.1%, 60.0%, and 69.2% patients, respectively, were in the TCE group. Of 1,656 patients in the non-transplant group, the proportion of cumulative TCE patients increased from 18.8% in the 2nd line to 21.2% in the 5th line or later. In the transplant group, among patients receiving 1st, 2nd, 3rd, 4th, and 5th or later lines of treatments, 21.1%, 38.2%, 50.0%, 66.7%, and 100% patients were in the TCE group. Of 128 patients in the transplant group, the proportion of cumulative TCE patients increased from 38.3% in the 2nd line to 43.8% in the 5th line or later.

The majority (89.7%) patients who were TCE in the 1st line had become so due to sequential therapies of multiple regimens, Vd-based to DRd-based sequencing being the most common (15.8%), followed by RVd-based to DRd-based (9.2%), and Rd-based to DVd-based (4.3%). One tenth (10.3%) of TCE patients had received a 1st-line four-drug combination regimen (daratumumab, lenalidomide, bortezomib, and dexamethasone). Among 184 non-transplant patients who were in the TCE group in the 1st line, 53 initiated treatment with daratumumab combination regimens (D-RVd-based, DVd-based, D-VMP-based, DRd-based).

Before approval of D-VMP for treatment of NDMM in Japan in August 2019, treatment with daratumumab combination regimens accounted for only 4% (3/73) patients; after its approval, the proportion increased markedly to 45% (50/111). After DRd approval (December 2019), daratumumab combination regimens accounted for most of the increase (51%, 49/96).

## Discussion

This retrospective study revealed real-world treatment trends and TCE status in each line in patients with NDMM who were eligible or not for SCT and who received daratumumab, lenalidomide, and/or bortezomib as 1st-line treatment in Japan. The study also clarified the characteristics and medical history of the strictly defined NDMM patients who were treated with these 1st-line drugs, and the preferred regimens for this group of patients by age and comorbidity. As there is limited evidence regarding treatment trends and treatment lines for these drugs in patients with NDMM, our findings on the real-world clinical practice in this setting would be of great benefit to clinicians now and in the future MM treatment landscape.

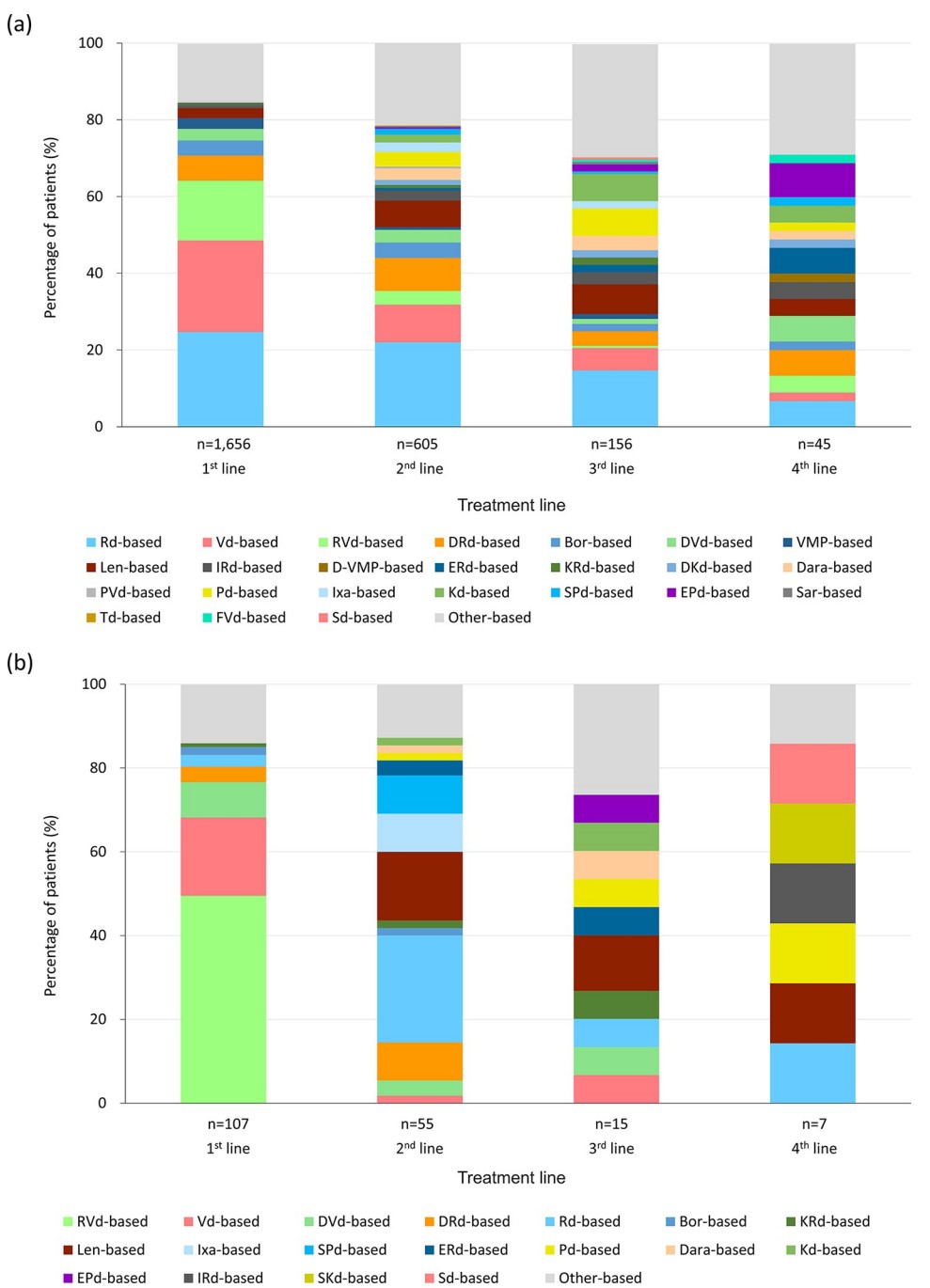

**Fig 2.** Treatment regimen in each line in (a) non-transplant and (b) transplant groups.

In this study, the majority of the patients in the non-transplant group were elderly (median age, 75 years; maximum, 94 years). In contrast, the transplant group had younger patients (median age, 61 years; maximum, 73 years). Previous studies conducted in Japan showed that SCT is usually used to treat younger MM patients without major comorbidities [10]. Though the JSH guidelines recommend SCT only in patients younger than 65 years without any serious comorbidities and with normal cardiopulmonary function [2], the guidelines also state

**Table 7. TCE patients.**

| Treatment line | Non-transplant group | | | Transplant group | | |
|---|---|---|---|---|---|---|
| | Total | TCE per line[a] | Cumulative TCE[b] | Total | TCE per line[a] | Cumulative TCE[b] |
| 1st line | 1,656 | 184 (11.1) | 184 (11.1) | 128 | 27 (21.1) | 27 (21.1) |
| 2nd line | 605 | 179 (29.6) | 312 (18.8) | 76 | 29 (38.2) | 49 (38.3) |
| 3rd line | 156 | 61 (39.1) | 337 (20.4) | 26 | 13 (50.0) | 52 (40.6) |
| 4th line | 45 | 27 (60.0) | 350 (21.1) | 9 | 6 (66.7) | 55 (43.0) |
| 5th line or more | 13 | 9 (69.2) | 351 (21.2) | 3 | 3 (100.0) | 56 (43.8) |

Data are presented as n (%).

[a]Calculated as percentage by using number of patients in corresponding treatment line as denominator.

[b]Calculated as percentage by using number of patients in the 1st treatment line as denominator.

TCE, triple-class exposed.

that "the age cutoff of 65 years is really only a guideline, and in practice the course of treatment is determined with consideration to biological age" [2]. This study showed that in actual clinical practice, SCT was performed in patients with MM aged >65 years, with minimal comorbidities and favorable physical condition.

In the non-transplant group, approximately half (48.5%) the patients received Rd- and Vd-based regimens in 1st line, suitable for patients who were ineligible for transplantation due to existing comorbidities or unfavorable physical condition. Use of other regimens increased in 2nd-line and later treatment, but no trend to use specific regimens was observed, although there was increased use of the DRd-based regimen, from 6.6% in the 1st line to 8.6% in the 2nd line. The addition of daratumumab to doublet regimens is considered acceptable for non-transplant patients, so the use of DRd and DVd regimens is expected to increase further. Use of Kd-based and pomalidomide and dexamethasone (Pd)-based regimens increased noticeably in 3rd-line treatment (7.1% each) from <4% in the 2nd line. Elotuzumab, pomalidomide, and

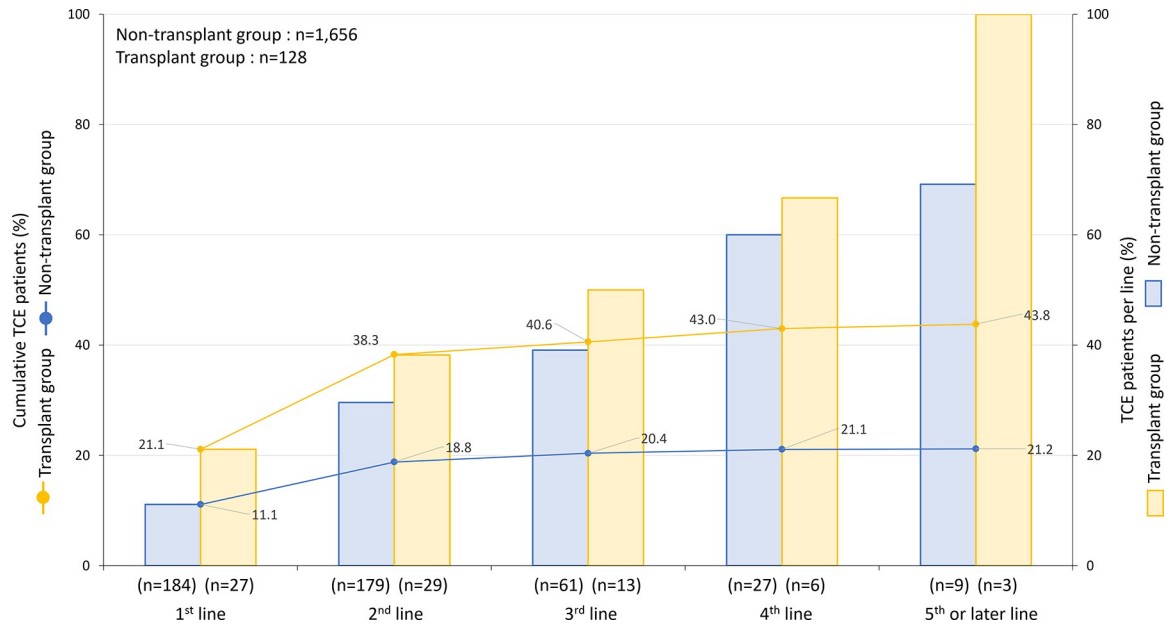

**Fig 3. Cumulative TCE patients and TCE patients per line.** TCE, Triple-class exposed.

dexamethasone (EPd)-based, and elotuzumab, lenalidomide, and dexamethasone (ERd)-based regimens were increasingly prescribed in the 4th line to 8.9% and 6.7% patients, respectively, versus <2% until 3rd-line treatment. The elotuzumab-containing triplet regimens, rather than Kd and Pd doublets, tended to be chosen in much later lines. It may be considered that elotuzumab has shown fewer side effects than other anti-myeloma agents [13].

In the transplant group, the RVd-based regimen was the most frequently prescribed, followed by Vd- and DVd-based regimens. On the other hand, bortezomib- and Rd-based regimens were prescribed as induction therapy for most patients prior to transplantation in 2nd-line treatment. In the 1st line, as induction therapy prior to SCT, regimens including bortezomib, especially triplet regimens, were preferentially selected for an anticipated rapid, deep response; however, due to the possible and less manageable severe side effect of peripheral neuropathy, they might rarely be selected in the 2nd or later lines after transplantation. For patients who received transplantation in the 2nd line, regimens containing carfilzomib (Kd or KRd) were preferentially selected versus those containing bortezomib, as these regimens are associated with higher response rates and lower incidence of peripheral neuropathy [14]. As this study showed that bortezomib or bortezomib-containing regimens (RVd, Vd, DVd) were used in 78.5% of transplant patients for 1st-line induction therapy (Table 6 and Fig 2), carfilzomib would be the preferable PI for 2nd-line induction therapy.

In the non-transplant group, the Rd-based regimen was preferred for patients >75 years of age, especially for those aged >84 years, while an RVd-based regimen was preferred for patients aged <65 years as reported previously [15], even if they were ineligible for transplantation. For patients aged 65–75 years, a Vd-based regimen was preferred over an RVd triplet regimen. In patients with renal dysfunction, a Vd-based regimen was preferred because of its effectiveness in improving renal dysfunction and no requirement for dose modification, regardless of creatinine clearance levels [16,17]. Patients with hepatic dysfunction were prescribed Rd or Vd, rather than RVd: this was also the case for patients with cardiac and vascular dysfunction. In agreement with previously published studies, the present study suggests that treatment pattern and regimen choice (doublet vs. triplet) should be based on the drugs' safety profiles, and patients' overall physical condition and the presence of comorbidities [18].

A higher proportion of patients in the transplant group moved to 2nd and subsequent lines than in the non-transplant group. The majority (63.5%) of non-transplant patients were treated in the 1st line only but more than half (59.4%) the transplant patients were prescribed 2nd or later lines. This group of transplant patients might have had a shorter duration of 1st-line treatment than non-transplant patients with subsequent treatment lines because they underwent four to six cycles of induction therapy followed by high-dose chemotherapy/autologous transplantation over a short period. More patients in the transplant group had to transit to 2nd- and 3rd-line treatments as their responses to 1st-line treatment were poor.

The longest duration of treatment (median: 18.96 months) was seen in transplant patients without subsequent-line therapy followed by non-transplant patients without subsequent-line therapy (median: 14.32 months). In contrast, transplant patients with subsequent-line therapy tended to show shorter duration of treatment than non-transplant patients in any lines, and the shortest duration of treatment (median: 2.92 months) was observed in transplant patients with subsequent-line therapy in the 2nd line. A large retrospective, cohort study conducted on 2,627 patients with NDMM by Corre et al. also showed that 18.9% patients with poor prognosis had early relapse within 18 months from the beginning of the 1st-line treatment and underwent SCT followed by subsequent lines of therapy [19]. Another retrospective study, conducted on 141 patients who had relapses before or 18 months after SCT, indicated that lack of maintenance therapy after initial treatment or after SCT could be one of the factors responsible for relapse within 18 months [20].

Our study showed that, of 128 patients in the transplant group, most (83.6%) underwent transplantation during 1st-line treatment, with 15.6% receiving transplantation during 2nd-line treatment. It is thought that these patients could have avoided transplantation, though they underwent stem cell collection after 1st-line induction therapy following a relapse, they underwent transplantation during the 2nd-line treatment [21]. Such patients may unfortunately relapse or become refractory in a short period after the onset of the 1st-line treatment and be considered patients with poor prognosis, because transplant generally show longer survival benefit. Therefore, transplantation might be selected as a better option in 2nd-line treatment.

Our results suggested that there was an increasing trend in the proportion of TCE patients, especially in transplant group patients (cumulative TCE rate; 21.2% and 43.8% in the non-transplant and transplant groups, respectively). Since DRd and D-VMP regimens have been integrated into 1st-line treatment, most patients are now TCE by the end of 2nd-line therapy. However, because the study period was from January 1, 2015 to December 31, 2022, the most patients in the study should have been TCE in later lines than the 2nd line. Most patients who received Vd-, RVd-, or Rd-based treatments transitioned to TCE after DRd-based, RVd-based, and DVd-based treatments. A proportion of patients received sequential treatment starting with a daratumumab-containing regimen in 1st-line treatment. After daratumumab approval for NDMM in 2019, and especially after DRd approval, there was a marked increase in the use of the daratumumab combination regimens. In both the non-transplant and transplant groups, most patients were TCE after 2nd- or later-line treatment, with RVd the most commonly used regimen in the transplant group. However, since there were far fewer patients in the transplant than the non-transplant group, further studies with larger numbers of patients are needed to confirm these findings.

This study has some limitations. First, the exclusion criteria that were set to identify accurately patients with NDMM—especially excluding patients with no anti-myeloma treatment records for at least 3 years before the index date—may have excluded patients unnecessarily. Second, in Japan, around 20% of NDMM patients are eligible for transplantation [5,22]. This study selected patients with NDMM with ≥6 months follow-up from the index date; therefore, patients with shorter follow-up would be excluded even if they were eligible for, or had received, transplantation, resulting in underestimation of the number of patients eligible for transplantation. Additional limitations are common to database studies: the MDV database contains claims data, rather than comprehensive medical and treatment histories of all patients as this information is not captured prior to database registration or outside registered hospitals, and is censored if the patient changes hospital. The reasons for treatment discontinuation, change of treatment line, data discontinuation, and administration status are unknown and the presence of comorbidities may be under- or overestimated because the MDV database holds only data of patients who required in-hospital tests. However, our comprehensive case definition that included the use of diagnostic codes, prescription details, and ≥6 months follow-up should mitigate any misclassification bias. Lastly, 15.1% and 14.0% of patients in the non-transplant and transplant groups, respectively, in the 1st line received 'other' regimens: These regimens were modified regimens, steroid uncombined regimens, or unapproved regimens. this may have led to underestimation of treatment regimens defined in each line.

## Conclusion

Our study showed that in the real-world setting, transplantation can be performed in patients with fewer comorbidities, aged up to 73 years. The present study showed that most transplant patients were hospitalized in large hospitals. In both non-transplant and transplant patients

receiving 1st-line treatment, most received an Rd-based or Vd-based doublet regimen, or an RVd-based triplet regimen. Most TCE patients had received DRd-based, RVd-based, or DVd-based regimens. The proportion of TCE patients showed an increasing trend after 2nd or later lines for both non-transplant and transplant patients; this was highest in the transplant group. The findings of this study will support the development of treatment strategies and policies for MM treatment in real-world clinical practice in Japan.

## Supporting information

**S1 Table. Comorbidities defined using the ICD10 classification in this study.**
(DOCX)

**S2 Table. Proportion of patients per treatment regimen by age in 1st line in the non-transplant group.**
(DOCX)

**S3 Table. Proportion of patients per treatment regimen by comorbidities in 1st line in the non-transplant group.**
(DOCX)

**S1 Fig. Study design.** MDV, Medical Data Vision.
(PDF)

**S2 Fig. Definitions of treatment lines and line transfer.**
(PDF)

**S3 Fig. Definitions of treatment lines and line transfer for transplant group.**
(PDF)

## Acknowledgments

We thank Dr. Kentaro Tajima of Pfizer Japan Inc. for providing critical advice for development of the study protocol and statistical analysis plan. The authors thank Niraj Vyas PhD, and Disha Dayal PhD, of MedPro Clinical Research for providing medical writing support for this manuscript.

## Author Contributions

**Conceptualization:** Toyoki Moribe, Linghua Xu, Kenshi Suzuki.

**Data curation:** Toyoki Moribe, Linghua Xu.

**Formal analysis:** Toyoki Moribe, Linghua Xu, Kazumi Take, Naohiro Yonemoto.

**Funding acquisition:** Toyoki Moribe.

**Investigation:** Toyoki Moribe, Linghua Xu.

**Methodology:** Toyoki Moribe, Linghua Xu, Kazumi Take, Naohiro Yonemoto, Kenshi Suzuki.

**Project administration:** Toyoki Moribe.

**Writing – original draft:** Toyoki Moribe.

**Writing – review & editing:** Toyoki Moribe, Linghua Xu, Kazumi Take, Naohiro Yonemoto, Kenshi Suzuki.

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
