## [Decision Letter · Decision Letter 0]

17 Jul 2024

PONE-D-24-24880Real-world treatment trends and triple class exposed status in newly diagnosed multiple myeloma patients in Japan: a retrospective claims database study.PLOS ONE

Dear Dr. Moribe,

Thank you for submitting your manuscript to PLOS ONE. After careful consideration, we feel that it has merit but does not fully meet PLOS ONE’s publication criteria as it currently stands. Therefore, we invite you to submit a revised version of the manuscript that addresses the points raised during the review process. The authors has to be state more clearly the manuscirpt and they could arrange their data more effectively.Some part is really hard to follow and cope.

We look forward to receiving your revised manuscript.

Kind regards,

Mehmet Baysal

Academic Editor

PLOS ONE

Journal Requirements:

    "I have read the journal's policy and the authors of this manuscript have the following competing interests: 

Toyoki Moribe, Linghua Xu, Kazumi Take, and Naohiro Yonemoto are employees of Pfizer Japan Inc., a sponsor of this study. Statistical analysis support was provided by Tatsuo Sakashita and Takayuki Sawada of Clinical Study Support, Inc. (Nagoya, Japan), which was funded by Pfizer Japan Inc."

We note that one or more of the authors are employed by a commercial company: Pfizer Japan Inc.

3. For studies involving third-party data, we encourage authors to share any data specific to their analyses that they can legally distribute. PLOS recognizes, however, that authors may be using third-party data they do not have the rights to share. When third-party data cannot be publicly shared, authors must provide all information necessary for interested researchers to apply to gain access to the data. (https://journals.plos.org/plosone/s/data-availability#loc-acceptable-data-access-restrictions) 

Reviewers' comments:

Reviewer's Responses to Questions

**Comments to the Author**

1. Is the manuscript technically sound, and do the data support the conclusions?

Reviewer #1: Partly

Reviewer #2: Yes

2. Has the statistical analysis been performed appropriately and rigorously? 

Reviewer #1: No

Reviewer #2: Yes

3. Have the authors made all data underlying the findings in their manuscript fully available?

Reviewer #1: Yes

Reviewer #2: Yes

4. Is the manuscript presented in an intelligible fashion and written in standard English?

Reviewer #1: No

Reviewer #2: Yes

5. Review Comments to the Author

Reviewer #1: Article title: 2 Real-world treatment trends and triple class exposed status in newly diagnosed multiple myeloma patients 3 in Japan: a retrospective claims database study.

In this manuscript, Moribe et al. describe the current treatment trends for patients with NDMM in Japan. The study does a good job of describing the regimens used across lines of therapy in both transplant-eligible and transplant-ineligible patients. It also describes the treatment trends according to underlying characteristics and comorbidities, which are useful for clinicians at the time of treatment selection. They also describe the percentage of triple-class exposed (TCE) patients across treatment lines. My main concerns are outlined below.

Title: does not match with the manuscript as a whole. The authors discuss NDMM and RRMM. Perhaps the authors should consider a more broad title like A retrospective study to define treatment patterns for patients in Japan with newly diagnosed and relapsed and refractory multiple myeloma.

Abstract: A bit all over the place.

• The authors need to comment as to why of the 1780 patients, why only 128 were considered transplant eligible. Why did they not include more transplant eligible patients?

• They majority of the abstract is focused on the NDMM patients and not RRMM.

• How is it that by the 5th line that only 21.2% and 43.8% of patients were triple class exposed?

Line 70: This sentence does not make any sense and should be rewritten.

Line 74: It is untrue that triple class exposed patients have limited treatment options. Almost all patients will have been triple class exposed by the end of 2nd line therapy (at the very latest) and most will be TCE during early induction. Are the authors considering TCE to be equivalent to triple class refractory (TCR). If so, this needs to be changed throughout the manuscript. Additionally, with the addition of CAR and bispecifics, treatment options are now considerably more than they were even a few years ago - this should be highlighted.

Line 114: It is important to highlight the time period in which patients had data collected and not just when the data was captured. The results suggest that patients treated with an initial diagnosis before 2015 were included. This is unclear.

Line 119-120 – Should read “ Patients were excluded from the study if they were prescribed anti-myeloma treatment drugs other than dexamethasone and prednisone before index date”

Line 124 – The definition of index date should be moved up to before the concept is first utilized (page 6 line 120). To make it easier for the reader.

Line 150 – Authors mentioned that the number of transfusions was a variable of interest, but this was never addressed.

Page 9; Study population – Although Figure 1 does a great job at explaining the breakdown of patients, the text is a little confusing and should be rewritten.

Line 188: Again, this is a very skewed population. Is autoHSCT not standarly used in Japan?

Line 198+: How are comorbidities defined. If this data is included, the authors need to explain what is meant by renal dysfunction, etc (e.g. CrCl between X and Y)

Table 4 - It is not clear to me what the difference between the cumulative number and proportion of patients using therapies across different treatment lines and the number and proportion of patients based on the most recent treatment lines being used is, this needs further explanation.

Table 5: This table is unclear and confusing. It is not clear how duration of treatment in 1st line treatment is broken down between patients prescribed and not prescribed subsequent treatment. Are you saying that duration of treatment to 1st line (non-transplant) in those patients treated with more than one line of therapy was 6.9 months vs 14.3 months in those patients that didn't get more than one line of treatment?

Table 6 - It would be more visually impactful if the table were reorganized to display the treatment regimens for the non-transplant and the transplant groups side by side rather than in sequential order, similar to Table 7.

"Frequently used treatment regimen": The first paragraph in this section needs clarity. The data is not presented in a way that is useful or informative. The table helps clarify this a bit but the authors should consider rewriting this section so that it is more clear. Additionally, in Table 6, the authors include IRd - has ixazomib been introducted anywhere in the manuscript? What about F, Sar, S, etc.? Consider moving all of the regimens that are not discussed in the body of the manuscript to the supplemental section.

Line 244: 24.7, 23.8, and 15.6 don't add up to 100%. What else were patients treated with? I thought the inclusion was that patients had to be treated with len, bort, or a combination of these.

Line 247: Not sure this makes sense. 24.7% of patients received Rd in 1st line and then 22% received Rd in the 2nd line. It would be more useful to understand the treatment pattern in my opinion. For example, did physicians change from Rd to Vd or Rd to RVd?

Line 261: Top 3? Perhaps use wording like the three more frequently prescribed.

Line 281: "in the TCE situation" should be rewritten

Table 7: Again, I am confused how any patients in the 4th or 5th line could not be triple class exposed. Also, with the attrition rate being so high, is cumulative TCE meaningful?

Line 294: This paragraph needs to be cleaned up, the description is not very clear though the information is important.

Line 315: Need to discuss why more transplant patients were not captured in this data set.

Lines 348-350 – The authors comment on utilizing K vs V in the second line mostly because of the difference in adverse events (namely peripheral neuropathy) – However, there should also be a mention of the difference in efficacy of K vs. V and also that as they clearly described nearly 50% of patients received bortezomib in the first line, making carfilzomib a better PI option in the second and further lines.

Line 357 onwards – The authors’ comment mentioned that “Lenalidomide containing regimens should be generally avoided because of their characteristic side effects, DVT and renal excretion pathway.” – This should be addressed as there is evidence that lenalidomide can be utilized in patients with renal dysfunction even in those undergoing HD as long as the dose is adjusted to the Creatinine Clearance. This statement might wrongly prevent providers from using lenalidomide, which, as we know, is a cornerstone of MM treatment in patients with any degree of renal dysfunction. Similarly, the authors also mentioned, “On the other hand, in patients with cardiac and pulmonary dysfunctions, Vd should be avoided because bortezomib was characterized by cardiac and pulmonary toxicities.” – In the ENDURANCE trial, only 5% of patients experienced at least one grade 2 or higher CV and pulmonary AE, and in most cases these were reversible. This statement could also wrongly lead clinicians to avoid bortezomib in patients with any degree of cardiopulmonary disease and should be modified.

Discussion: Much too long, this needs to be shortened and written more concisely. A segment focusing on proposing the standardization of quadruplet therapy in NDMM should be added – if this aligns with the resources and available drug approval in Japan.

Conclusion: Sentences 1 and 2 in this paragraph are not the reason for this study and this data was not highlighted in the manuscript, so why do the authors include it here?

Reviewer #2: Thanks to the authors for the article. A useful study reflecting real life data. The study nicely discussed the treatments applied to patients diagnosed with multiple myeloma in Japan. I am just wondering why there is a significant difference in the number of transplant-ineligible and transplant-eligible patients.

6. PLOS authors have the option to publish the peer review history of their article (what does this mean?). If published, this will include your full peer review and any attached files.

Reviewer #1: No

Reviewer #2: No

---

## [Author Response · Author response to Decision Letter 0]

12 Aug 2024

Reviewers' comments:

Reviewer's Responses to Questions

Comments to the Author

Thank you for your kind overall evaluation and your valuable time spent on reviewing our manuscript. We have answered your comments below and revised the manuscript accordingly. In addition, we have also revised according to comments from reviewers and made some corrections based on fact-check.

1. Is the manuscript technically sound, and do the data support the conclusions?

Reviewer #1: Partly

Reviewer #2: Yes

Response:

We have made minor revisions of the conclusion based on the data presented in this study as below. 

“Our study showed that in the real-world setting, transplantation can be performed in patients with fewer comorbidities, aged up to 73 years. The present study showed that most transplant patients were hospitalized in large hospitals. In both non-transplant and transplant patients receiving 1st-line treatment, most received an Rd-based or Vd-based doublet regimen, or an RVd-based triplet regimen. Most TCE patients had received DRd-based, RVd-based, or DVd-based regimens. The proportion of TCE patients showed an increasing trend after 2nd or later lines for both non-transplant and transplant patients; this was highest in the transplant group. The findings of this study will support the development of treatment strategies and policies for MM treatment in real-world clinical practice in Japan.” (Page 27 of the track file of the revised manuscript)

2. Has the statistical analysis been performed appropriately and rigorously? 

Reviewer #1: No

Reviewer #2: Yes

Response:

In this study the data were summarized descriptively. The patient characteristics, duration of treatment, treatment pattern, and other variables in transplanted and non-transplanted patients who are newly diagnosed with MM were evaluated descriptively with summary statistics. This set of analyses is intended mainly for descriptive purposes and to provide a basis for future studies. P-values were not calculated to investigate any prespecified set of hypotheses. Further, all data analysis was performed using statistical software SAS version 9.4 (SAS Institute, Cary, NC, USA). We have already included all these details in the statistical analysis section in the manuscript file. (Last paragraph of statistical analysis section on Page 9 of the track file of the revised manuscript)

3. Have the authors made all data underlying the findings in their manuscript fully available?

Reviewer #1: Yes

Reviewer #2: Yes

Response:

Thank you for your review and confirmation.

4. Is the manuscript presented in an intelligible fashion and written in standard English?

Reviewer #1: No

Reviewer #2: Yes

Response:

We have carefully looked over the manuscript and made some corrections for the language and consistency as below. We have got the manuscript edited by a native English editor as well and editor has made editorial changes throughout the manuscript. These changes are shown as track changes in the track version of the revised manuscript file.

5. Review Comments to the Author

Reviewer #1: 

Article title: Real-world treatment trends and triple class exposed status in newly diagnosed multiple myeloma patients in Japan: a retrospective claims database study.

In this manuscript, Moribe et al. describe the current treatment trends for patients with NDMM in Japan. The study does a good job of describing the regimens used across lines of therapy in both transplant-eligible and transplant-ineligible patients. It also describes the treatment trends according to underlying characteristics and comorbidities, which are useful for clinicians at the time of treatment selection. They also describe the percentage of triple-class exposed (TCE) patients across treatment lines. My main concerns are outlined below.

Response:

We greatly appreciate your positive feedback. Moreover, we also thank you for your kind advice and review comments. Here we are providing a point-by-point response. 

1) Title: does not match with the manuscript as a whole. The authors discuss NDMM and RRMM. Perhaps the authors should consider a more broad title like A retrospective study to define treatment patterns for patients in Japan with newly diagnosed and relapsed and refractory multiple myeloma.

Response:

Thank you for your suggestion. As you pointed out, the results include data for both NDMM and RRMM patients. However, this study mainly focused on NDMM patients rather than RRMM patients after strict definition of NDMM patients. Moreover, as for RRMM setting, the data has just been presented as the treatment trend and TCE status among NDMM patients receiving 2nd, 3rd, 4th, and 5th or later lines of treatments. Therefore, we would like to keep the title unchanged.

2) Abstract: A bit all over the place.

• The authors need to comment as to why of the 1780 patients, why only 128 were considered transplant eligible. Why did they not include more transplant eligible patients?

• They majority of the abstract is focused on the NDMM patients and not RRMM.

• How is it that by the 5th line that only 21.2% and 43.8% of patients were triple class exposed?

Response:

• This is a limitation of this study. As Japanese patients with MM are very elderly (with median age >75 years) and have multiple comorbidities, only 20% of NDMM patients are eligible for transplantation. Furthermore, in this study, NDMM patients who had a follow-up period of ≥6 months from the index date have been selected. When the patients had shorter follow-up period, the medical record for transplantation might not be detected even if the patients have received transplantation after the follow-up period in this study. Therefore, the number of transplant eligible patients may be underestimated.

• As you pointed out, this study focused mainly on NDMM patients rather than RRMM patients.

• The NDMM patients have been first identified according to inclusion and exclusion criteria and then evaluated the TCE situation in each treatment line for the patients. However, because the follow-up period was ≥6 months from the index date, more than half of the NDMM patients remained in the 2nd line. Just a few patients transferred up to the 5th line. Thus, the MM patient number has gradually become less from the 1st line to the 5th line, though the TCE patient number has gradually increased from the 1st line to the 5th line. Therefore, the cumulative TCE patients in the 5th line may be underestimated.

3) Line 70: This sentence does not make any sense and should be rewritten.

Response:

According to your comment, we have rewritten the sentence as below.

“However, daratumumab combination regimens used in the 1st line have complicated MM treatment and fragmented later-line MM treatment sequences.” (Last three lines of first paragraph on Page 4 of the track file of the revised manuscript)

4) Line 74: It is untrue that triple class exposed patients have limited treatment options. Almost all patients will have been triple class exposed by the end of 2nd line therapy (at the very latest) and most will be TCE during early induction. Are the authors considering TCE to be equivalent to triple class refractory (TCR). If so, this needs to be changed throughout the manuscript. Additionally, with the addition of CAR and bispecifics, treatment options are now considerably more than they were even a few years ago - this should be highlighted.

Response:

As you pointed out, most patients will have been TCE by the end of 2nd line therapy most recently, since DRd and D-VMP regimens have been integrated in the 1st line. However, because the study period was from January 1, 2015 to December 31, 2022, the most patients in this study should have been TCE in later lines than the 2nd line.

We have added this information into the description as below:

“Our results suggested that there was an increasing trend in the proportion of TCE patients, especially in transplant group patients (cumulative TCE rate; 21.2% and 43.8% in the non-transplant and transplant groups, respectively). Since DRd and D-VMP regimens have been integrated into 1st-line treatment, most patients are now TCE by the end of 2nd-line therapy. However, because the study period was from January 1, 2015 to December 31, 2022, the most patients in the study should have been TCE in later lines than the 2nd line.” (Second paragraph on Page 25 of the track file of the revised manuscript)

Further, we consider TCE is not equivalent to TCR. Most patients with MM who are TCE will eventually have relapse and TCE patients who have relapsed or are refractory may have a poor prognosis with worsening outcomes and limited treatment options in clinical practice considering the physical condition of patients. Below are the references: 

• Gandhi UH, Cornell RF, Lakshman A et al. Outcomes of patients with multiple myeloma refractory to CD38-targeted monoclonal antibody therapy. Leukemia 33(9), 2266–2275 (2019).

• Kumar SK, Dimopoulos MA, Kastritis E et al. Natural history of relapsed myeloma, refractory to immunomodulatory drugs and proteasome inhibitors: a multicenter IMWG study. Leukemia 31(11), 2443–2448 (2017).

• Nijhof IS, van De Donk N, Zweegman S, Lokhorst HM. Current and new therapeutic strategies for relapsed and refractory multiple myeloma: an update. Drugs 78(1), 19–37 (2018).

• Mikhael J. Treatment options for triple-class refractory multiple myeloma. Clin. Lymphoma Myeloma Leuk. 20(1), 1–7 (2020).

We agree with your comment “with the addition of CAR and bispecifics, treatment options are now considerably more than they were even a few years ago”. According to your comment, we revised the description as below.

“TCE patients have limited treatment options, especially before chimeric antigen receptor T-cell (CAR-T) therapies and bispecific antibodies (BsAbs) become available.” (Second paragraph on Page 4 of the track file of the revised manuscript)

5) Line 114: It is important to highlight the time period in which patients had data collected and not just when the data was captured. The results suggest that patients treated with an initial diagnosis before 2015 were included. This is unclear.

Response:

In this study, the results do not include the data for patients treated with an initial diagnosis before 2015 (before the study period), because we focused the time period after lenalidomide has become available for NDMM patients in Japan. To clearly define incident NDMM diagnosis in the study period in this study, we excluded patients who did not have any medical record for at least for 3 years or had received anti-myeloma treatments and MM diagnosis before the index date. Moreover, even if we changed the window period of 3 years (to 2 years, 1 year and 6 months), the patient characteristics did not change materially, suggesting minimal possibility of bias. 

We added description to clarify this point as below.

“...Multiple myeloma and malignant plasma cell neoplasms), had ≥6 months follow-up data, and were prescribed bortezomib, lenalidomide, and/or daratumumab after the 1st diagnosis of MM were included in the study. Therefore, the results did not include the data of treated patients with an initial diagnosis before 2015.” (First paragraph of ‘Study population and study period’ on Page 6 of the track file of the revised manuscript)

6) Line 119-120 – Should read “ Patients were excluded from the study if they were prescribed anti-myeloma treatment drugs other than dexamethasone and prednisone before index date”

Line 124 – The definition of index date should be moved up to before the concept is first utilized (page 6 line 120). To make it easier for the reader.

Response:

We agree with your suggestion. We moved the definition of index date up just after explaining the study period and inclusion criteria.

Therefore, the results do not include the data of treated patients with an initial diagnosis before 2015. The index date was defined as the earliest prescription date of the 1st-line drugs in the same month or months after the first diagnosis of MM. (First paragraph of ‘Study population and study period’ on Page 6 of the track file of the revised manuscript)

7) Line 150 – Authors mentioned that the number of transfusions was a variable of interest, but this was never addressed.

Response:

We apologize for this error. We deleted transfusion and corrected this description as follows:

“Specific outcome variable of interest was TCE status in each line.” (Paragraph above ’Statistical analysis’ on Page 8 of the track file of the revised manuscript)

8) Page 9; Study population – Although Figure 1 does a great job at explaining the breakdown of patients, the text is a little confusing and should be rewritten.

Response:

We have rewritten the text to explain the study population as below.

Over the study period, 44,897 patients had a confirmed diagnosis of MM, of which 16,021 patients had prescriptions for daratumumab, lenalidomide, and/or bortezomib at the index date (Fig 1). Of 12,255 patients who had a follow-up period of at least 6 months from the index date, 10,471 patients who met the other three exclusion criteria—were prescribed anti-myeloma treatment drugs other than dexamethasone and prednisone before the index date , had no medical records before the index date for ≥3 years, and received SCT before the index date—were excluded. Thus, 1,784 patients were included in the final cohort analysis set. Of the 1,784 patients, 1,656 patients were transplant ineligible (non-transplant group) and 128 patients were transplant eligible (transplant group) (Fig 1). (‘Study population’ paragraph on Page 10 of the track file of the revised manuscript)

9) Line 188: Again, this is a very skewed population. Is autoHSCT not standarly used in Japan?

Response:

This is a limitation of this study. As Japanese patients with MM are elderly (with median age >75 years) and have multiple comorbidities, around 20% of NDMM patients are eligible for transplantation. Furthermore, in this study, NDMM patients who had a follow-up period of at least 6 months from the index date have been selected. When the patients had shorter follow-up period, the medical record for transplantation might not be detected even if the patients have received transplantation. Therefore, the number of transplant eligible patients may be underestimated.

We have added this as limitation as below.

Second, in Japan, around 20% of NDMM patients are eligible for transplantation [5,22]. This study selected patients with NDMM with ≥6 months follow-up from the index date; therefore, patients with shorter follow-up would be excluded even if they were eligible

---

## [Decision Letter · Decision Letter 1]

29 Aug 2024

Real-world treatment trends and triple class exposed status in newly diagnosed multiple myeloma patients in Japan: a retrospective claims database study.

PONE-D-24-24880R1

Dear Dr. Moribe,

We’re pleased to inform you that your manuscript has been judged scientifically suitable for publication and will be formally accepted for publication once it meets all outstanding technical requirements.

Kind regards,

Mehmet Baysal

Academic Editor

PLOS ONE

Additional Editor Comments (optional):

Reviewers' comments:

Reviewer's Responses to Questions

**Comments to the Author**

1. If the authors have adequately addressed your comments raised in a previous round of review and you feel that this manuscript is now acceptable for publication, you may indicate that here to bypass the “Comments to the Author” section, enter your conflict of interest statement in the “Confidential to Editor” section, and submit your "Accept" recommendation.

Reviewer #2: All comments have been addressed

2. Is the manuscript technically sound, and do the data support the conclusions?

Reviewer #2: Yes

3. Has the statistical analysis been performed appropriately and rigorously? 

Reviewer #2: Yes

4. Have the authors made all data underlying the findings in their manuscript fully available?

Reviewer #2: Yes

5. Is the manuscript presented in an intelligible fashion and written in standard English?

Reviewer #2: Yes

6. Review Comments to the Author

Reviewer #2: I have no more additional comments. Thanks to the authors. I think this study will provide interesting data on the approach to MM in Japan.

7. PLOS authors have the option to publish the peer review history of their article (what does this mean?). If published, this will include your full peer review and any attached files.

Reviewer #2: No

---

## [Editor Report · Acceptance letter]

22 Sep 2024

PONE-D-24-24880R1 

PLOS ONE

Dear Dr. Moribe, 

I'm pleased to inform you that your manuscript has been deemed suitable for publication in PLOS ONE. Congratulations! Your manuscript is now being handed over to our production team.

Kind regards, 

on behalf of

Dr. Mehmet Baysal 

Academic Editor

PLOS ONE